# Online Practicum Effectiveness: Investigation on Tutors' and Student-Teachers' Perceptions Pre/Post-Pandemic

**Loredana Perla [1,*] and Laura Sara Agrati [2,*]**

1 Dipartimento di Scienze Umane e Sociali, University of Bari, 70121 Bari, Italy
2 Dipartimento Di Scienze Della Formazione, Psicologia, Comunicazione, University of Bergamo, 24129 Bergamo, Italy
* Correspondence: loredana.perla@uniba.it (L.P.); laurasara.agrati@unibg.it (L.S.A.)

**Abstract:** The practicum is an essential activity for initial teacher training courses as it ensures the immersion in real contexts and sharing practice with experts. The early experiments on online practicums occurred thanks to the development of technologies and in view of overcoming unbridge-able distances. In the COVID-19 pandemic phase, online practicum experiences are carried out and studies have investigated their effects in general and on the student-teachers involved. This work presents the first results of an investigation on the perceptions of effectiveness of 578 student-teachers and 33 tutors regarding the online practicum carried out at the 'Primary Education Sciences' bachelor's degree at the University of Bari and Bergamo. Quantitative and qualitative data were collected through a mixed questionnaire and analyzed statistically and through the QDA procedure. The study offers elements to understand the effectiveness of the online practicum from the perspective of student-teachers and tutors, i.e., overcoming spatial distance and greater sharing of experiences and materials. It paints online practicum as a positive post-pandemic reality and offers some useful insights for teacher training course leaders and teacher education scholars.

**Keywords:** online practicum; teacher education; tutoring

## 1. Introduction—Necessity and Challenges in the Online Practicum

The practicum, in terms of practical training activity, is a necessary component within initial teacher training courses since it allows novice teachers to immerse themselves in real-life contexts and to share the practices of experienced teachers [1–3]. Today's professional learning paradigm (O'Brien & Jones, 2014) and the resulting increased recognition of the professional environment as a complex and interacting system of 'activity networks' clarified the value of the practicum for the purposes of preparing teachers. In the practicum, particular 'micropolitical' conditions, essential for the growth especially of novices—e.g., formal accompaniment support in the profession (cf. mentoring), support network from expert colleagues (cf. induction)—would be realized [4,5]. Also for this, from the perspective of professional learning, the teaching practicum continues to be considered the most critical period for trainee teachers as they have the opportunity to verify their personal knowledge, skills and intentions within real classroom scenarios and in the actual dynamics of the school and to feel that sense of inadequacy [6,7], due to differences in what they live in relation to their direct experience and what they have been taught in their program of initial teacher education.

In teacher training carried out in the higher education segment [8], the challenge for academies is to train students' practical knowledge, based on the close connection between the theoretical and experiential component of the programs, through the formative alternation between lectures, laboratories and practicum, at the pedagogical level, and through the collaboration between professors, expert tutors and students, at the operational level.

Due to the COVID-19 pandemic, the higher education has forcefully entered the era of distance learning [9], by experimenting with new solutions aimed at ensure the

development of students' competences—i.e., e-learning environments where to adapt learning activities are no longer feasible face to face [10,11].

Through more sophisticated technologies—e.g., platforms for visualization and sharing the materials [12]—it was possible to recreate experience-based activities, which have always been carried out in person, in online mode. However, to encourage the exchange and fruitful collaboration between professors, experts and students, it was, however, necessary to ensure access to devices, to reorganize the curriculum both at the instructional and learning levels, in a hybrid [13] and augmented [14] sense, as well as to rethink communication and teaching-learning practices [15,16].

The disruption of the practicum during the COVID-19 pandemic in qualifications-related programs has rapidly increased the scope of virtualization [17], systematically incorporated into curriculum redesign [18]. Some studies tried to balance the challenges and opportunities offered by the practicum with reference to the pandemic phase in which adaptation to the online mode was experienced as a necessity [12,19–22].

The challenge most felt by university teachers, student-teachers and expert tutors was the lack of vivid communication and interaction between pre-service teachers and students during online lessons to the point of thinking that it was going to undermine the pre-service teachers' professional development, namely social and personal [12,22]. No less important is the lack of personal support between student-teacher and mentor [12,23] regarding immediate feedback. Another challenge was considered the lack of experience and expertise in utilizing technology and digital materials [24]—e.g., the difficulty to transpose online the representation of concrete objects or situations that would activate student-teacher learning was felt [22].

Furthermore, the use of asynchronous and synchronous platforms once seen as having potential, e.g., unlimited access due to time flexibility [20], is now seen as a challenge or even a limitation, e.g., overcoming the barrier between private and public life.

*Perception of Effectiveness about the Online Practicum*

After the pandemic crisis, 'beyond the covid era' [9], significant experiences and studies remain that open up new possibilities for reading online practices. Studies found that, in general, the main opportunities associated with the online practicum were as follows:

- Familiarity with technologies (software and online platforms) and the possibility of increasing technological skills in an operational way and directly connected to their use [12]
- Most frequent and successful interaction between student-teachers, professors and expert teachers during online teaching practicum, and also in terms of communicative, emotional and technical support [19];
- Immediate feedback and constructive support (focused on the work produced and the actions carried out) between student-teachers and university professor, without having to travel and wait to meet [19];
- Mutual feedback between student-teachers and mentor since the former, in a virtual teaching environment, have the possibility to better observe the actions of the expert [20];
- Most reflection on the socio-cultural environment of students and their families during the pandemic and its impact on learning [12]—e.g., better understanding of the impossibility/difficulty of access and the risks of exclusion; ability to involve students not supported by the family network, etc.

The study by Brina and Psoni [25], specifically, investigates whether the online practicum is effective with respect to the skills of student-teachers even when it is no longer necessary. Qualitative analyzes show positive effects on the use of new technologies in education as well as on transversal skills such as adaptability, flexibility and management of student interaction in online contexts. The study by Giner-Gomis and colleagues [26] also investigated the teaching practicum during COVID-19 through the lens of pre-service

teachers, noting instead that the general quality of initial training has decreased due to the adjustments and adaptations implemented in school contexts.

The study by Cutri, Mena and Feinauer-Whiting [16] focused attention on the longitudinal perceptions of preparation and readiness of faculty teachers and brought to light that, net of access difficulties, the negative connotations of risk taking and making mistakes while teaching online seem to have been mitigated by affective factors, such as empathy and optimism, related in some way to the unexpected advantages regarding collaboration with students given by the more flexible times. The research by Hergan and Pečar [27]—which focused, more specifically, on the evaluation by primary school teacher students of the usefulness of their distance teaching practicum during COVID-19—confirms the importance of flexible times (albeit with a significant gap between younger and older trainees). Indeed, the survey highlighted that the different timing mainly made it possible to prepare the teaching materials more accurately and to offer individual help to the students.

## 2. Investigation on the 'Primary Education Sciences' Degree Course in UniBa and UniBg

The investigation accomplished between January 2022 and April 2023 in two Italian universities (University of Bari and University of Bergamo) focuses on the hybrid organizational–didactic model [13] emerging from the adaptation of the practicum activities within the qualifying degree course for primary school teachers.

### 2.1. The Purpose of the Study

The investigation deepened the perceptions and representations of tutors and student-teachers regarding the methods of providing the practicum activities (face to face and online) of the degree course in 'Science of primary education' before, during and after the pandemic phase. Specifically, the study aimed to know the point of view of tutors and student-teachers regarding the remote mode—of the online and synchronous type—which has sometimes been resorted to in an emergency form during and post the COVID-19 pandemic phase. In this way, useful elements to learn about online practicum experiences and related representations by student-teachers and tutors were obtained [25,26].

The study reported analyzes, in particular, the opinions of tutors and student-teachers regarding the participation and types of online practicum activities and the perception of effectiveness. This paper addresses the following research questions with respect to online practicum activities:

(1) *What participation was there in the pre-, during- and post-pandemic phases?*
(2) *What types of practicum activities were delivered online from May 2021 to today?*
(3) *What perception of effectiveness tutors and student-teachers have?*

These questions were investigated through the analysis of answers nos. Q6–Q16 and questionnaire section—cf. Table 2.

### 2.2. Context

The 'Primary Education Sciences' bachelor's degree in Italy facilitates the professional development of primary school and kindergarten teachers (class LM85bis, Minister Decree n. 249/2010). The path lasts 5 years and provides at least 24 university credits (equal to 600 h) of practicum activities, starting from the second year, as explained in Table 1.

The general purpose of the practicum activities in the 'Primary Education Sciences' bachelor's degree is to foster in students the professional awareness relating to the specificities of nursery and primary schools and the development of personal skills oriented towards professional action in situations. Such practicum activities are divided into a real practicum at school and reflection meetings, held at the University, on the experience carried out at school during the practicum. Attendance is compulsory, both for practicum and reflection meetings. Such practicum activities are accompanied by three tutoring roles:

- School tutor (*mentor*)—a qualified expert teacher, appointed by the school principal, who welcomes trainee students into their practicum activities at schools;

- University tutor (*tutor*)—a qualified expert, selected by the university, who accompanies trainee students in the processes of critical reflection on the practices observed/acted at school and in the processes of self-assessment and evaluation of the training course undertaken;
- Organizer tutor (*supervisors*)—usually a school principal, who maintains contact with the welcoming school while also representing an institutional point of view and the envisaged authorizations, and coordinates the interventions of the university tutors.

**Table 1.** Practicum hours per year at the bachelor's degree in 'Primary Education Sciences'.

| Year of Study | Hours/n. Universities Credits | Practicum (at School) | Reflection on Practicum (at the University) |
|---|---|---|---|
| II | 100 (4) | 60 | 40 |
| III | 150 (6) | 100 | 50 |
| IV | 150 (6) | 100 | 50 |
| V | 200 (8) | 150 | 50 |
| Tot. | 600 (24) | 410 | 190 |

The presidential decree of 9 March 2020 determined the closure of schools in Italy and, therefore, also the practicum activities for teacher training. The ministerial decree n. 61 (5 May 2020) thereafter allowed schools, 'very exceptionally', to carry out practicum activities remotely at school, only as a support to teaching activities. The ministerial note of September 2020 then made it possible to return to carrying out the practicum at school, in compliance with the anti-COVID-19 safety protocols. In this way, in Italy, it was possible to carry out the practicum activities in the usual way in the first half of the 2019/20 academic year and in the second half of the 2020/21 academic year. In the intermediate phase—i.e., between the second half of the 2019/20 academic year and the first half of the 2020/21 academic year—however, exceptional experiments were carried out mainly focused on reflection on practicum.

*2.3. Collection Tool*

The data were collected through a mixed questionnaire that includes closed and open-ended questions concerning the following main sections: a. sociometric-professional information; b. pre- and during-COVID-19 online practicum activities; c. post-COVID-19 online practicum activities; d. perception of effectiveness about online practicum activities experienced; e. personal consideration about online practicum activities; f. report of experience (Table 2). The questionnaire was sent via e-mail to student-teachers and tutors in two regions of Italy—Puglia and Lombardy. The mixed collection tool facilitated, at the same time, the collection of quantitative and qualitative data subjected to triangulation to decide which type was more likely to provide the desired information [28].

**Table 2.** Questionnaire sections, questions and data types.

| | *Section* | *Information* (n. Question) | *Data* |
|---|---|---|---|
| a. | socio-metric and professional information | age (Q1), gender (Q2), qualification (Q3), length of service (only tutor) (Q4), enrollment year (only student-teacher) (Q5) | quantitative |
| b. | pre- and during-COVID-19 online practicum activities | participation in online practicum activities pre-COVID-19 (Q6), during COVID-19 (Q7) | quantitative |

**Table 2.** *Cont.*

|  | *Section* | *Information* **(n. Question)** | *Data* |
|---|---|---|---|
| c. | post-COVID-19 online practicum activities | participation in online practicum activities (Q8), types of activities (Q9), hours (Q10), mode of conduct (Q11) | quantitative |
| d. | perception of effectiveness about online practicum activities experienced | perception of effectiveness in general (Q12), with respect to face-to-face activities (Q13), based on types ('practicum'/'reflection on practicum') (Q14) | quantitative |
| e. | personal consideration about online practicum activities | strength (Q15) and weakness (Q16) | quantitative |
| f. | report of experience | description of an example online practicum activity | qualitative |

### 2.4. Participants and Data Analysis

The questionnaire was administered between January 2022 and April 2023 using a random criterion. A total of 611 responses were collected: 33 responses from tutors (n. 27 coordinators, n. 6 supervisors—5.6%) and 578 from student-teachers (94.4%). The following table shows the professional socio-metric information—see Table 3:

**Table 3.** Socio-metric and professional information.

| Information | Tutor (n. 33) | Student-Teacher (n. 578) | Tot. 611 |
|---|---|---|---|
| Age | 53.7 m | 26.2 m | 39.9 m |
| Gender | f (31, 94%) m (2, 6%) | f (553, 95.7%) m (25, 4.3%) | f (584, 95.6%) m (27, 4.4%) |
| Qualification | Diploma (5, 15.1%) Master's degree (23, 69.7%) PhD (5, 15.1%) | Diploma (423, 73.2%) Master's degree (153, 26.5%) PhD (2, 0.4%) | Diploma (428, 70.1%) Master's degree (176, 28.8%) PhD (7, 1.1%) |
| Length of service | 3.4 m | | |
| Enrollment year | | II (157, 27.1%) III (139, 24%) IV (133, 23.1%) V (148, 25.7%) | |

The population involved was 40 years old (39.9), largely female (95.6%) and mainly graduated (70.1%). The segment of tutors is older (53.72), female (94%), mostly graduates (69.7%) and having 3.4 years of service. The student-teacher segment was 26.2 years old, female (95.7%), having a high school diploma (73.2%) and fairly equally enrolled in the years in which the practicum is expected.

The quantitative data were managed based on a twofold statistical analysis: descriptive, to express the absolute values/percentages of the answers and the averages; inferential, to express the correlation between the descriptive evidence that emerged and some socio-metric and professional characteristics. This work reports the first results of the statistical analysis conducted on quantitative data.

Quantitative data were analyzed using SPSS (19.0). The treatment of textual material of the open answer took place through a qualitative data analysis [28]. Specifically, coding was articulated in three phases: a. open coding, as conceptualization through meaningful

text units and the identification of labels; b. axial coding, as the identification of frequent macro-categories emerging from the text units, along with the number of occurrences; c. selective coding, as hierarchical and analytical ordering of the identified macro-categories, for the final emergence of core categories.

## 3. Findings

The results are reported below in response to the research questions and summarized at the end of the paragraph. The findings refer to the absolute values and % related the population involved.

(1)    *What participation was there in the pre-, during- and post-pandemic phases?*

The following table (see Table 4) shows the absolute data and % of the answers to the questions on participation in online practicum activities on three phases—pre-, during- and post-COVID-19 (see Q6, Q7 and Q8).

**Table 4.** Participation in online practicum activities. Pre-, during- and post-COVID-19 comparison.

| Question: | Tutor (n. 33) | Student-Teacher (n. 578) | Tot. 611 |
|---|---|---|---|
| Q6 pre-COVID-19 (until March 2020) | | | |
| yes | 13 (39.4%) | 126 (21.8%) | 139 (22.7%) |
| no | 18 (54.5%) | 440 (76.2%) | 458 (75%) |
| other | 2 (6%) | 12 (2%) | 14 (2.3%) |
| Q7 during-COVID-19 (from March 2020 to May 2021) | | | |
| yes | 25 (75.7%) | 364 (63%) | 389 (63.7%) |
| no | 6 (18.2%) | 203 (35.1%) | 209 (34.2%) |
| other | 2 (6%) | 11 (1.9%) | 13 (2.1%) |
| Q8 post-COVID-19 (from May 2021) | | | |
| yes | 26 (78.7%) | 426 (70.7%) | 452 (74%) |
| no | 7 (21.2%) | 152 (26.3%) | 159 (26%) |
| other | 0 (0%) | 0 (0%) | (0, 0%) |

In general, increased participation in online practicum activities on the transition from the pre-COVID-19 phase (22.7%) to the during-COVID-19 phase (63.7%) to the post- phase (74%) of the COVID-19 pandemic, up to a complete reversal of relationships, is highlighted. In particular, the following can be noted:

- With reference to the during-COVID-19 phase, only a quarter of the tutors did not participate in online practicum activities (18.2%), compared to a third of the student-teachers (35.1%);
- With reference to the post-COVID-19 phase, a third of tutors (21.2%) and a third of student-teachers (26.3%) did not participate in online practicum activities.

(2)    *What types of practicum activities were delivered online from May 2021 to today?*

With reference to the online practicum activities carried out from May 2021 to April 2023, the following table (see Table 5) shows the absolute data and % of the answers to the questions on types of activities, hours and mode of conduct (see Q9, Q10 and Q11).

The prevalent types of online practicum activities carried out in the last period starting from May 2021 were a reflection on practicum (83%), lasting around 12 h per year and conducted in the plenary assembly mode (72.1%), followed by work in small groups (19.6%). From the comparison between the answers of the tutors and the student-teachers, a constant percentage emerges.

(3)    *What perception of effectiveness do tutors and student-teachers have of online practicum activities?*

The following table (Table 6) shows the absolute data and % of the answers to the questions on the tutors' and student-teachers' perception of effectiveness on online practicum activities ('reflection on practicum') experienced in the last period starting from May 2021. Perception is expressed in general (Q13), in relation to the face-to-face activities (Q14) and focusing on strength and weakness (Q18 and 19).

**Table 5.** Types of activities, hours and mode of conduct for online practicum activities experienced—from May 2021 to April 2023.

| **Question:** | | | |
|---|---|---|---|
| Q9—Types of activities | Tutor (n. 26) | Student-teacher (n. 440) | Tot. 466 |
| practicum (school) | 0 (0%) | 4 (0.9%) | 4 (0.9%) |
| reflection on practicum (univ.) | 22 (84.6%) | 365 (82.9%) | 387 (83%) |
| both | 4 (15.3%) | 71 (16.2%) | 75 (15.0%) |
| Q10—Hours | Tutor (n. 26) | Student-teacher (n. 522) | Tot. 548 |
| | 9.4 | 12.7 | 11.5 |
| Q11—Mode of conduct | Tutor (n. 26) | Student-teacher (n. 465) | Tot. 491 |
| plenary | 12 (46.2%) | 342 (73.5%) | 354 (72.1%) |
| small group | 7 (26.9%) | 89 (19.2%) | 96 (19.6%) |
| individual | 5 (19.2%) | 7 (1.5%) | 12 (2.4%) |
| other | 2 (7.7%) | 27 (5.8%) | 29 (5.9%) |

**Table 6.** Perceptions of effectiveness of online practicum activities, in general and related those experienced.

| **Question:** | | | |
|---|---|---|---|
| Q12—In general | Tutor (n. 33) | Student-teacher (n. 578) | Tot. 611 |
| not at all | 4 (12.1%) | 25 (4.3%) | 29 (4.7%) |
| little | 3 (9%) | 47 (8.2%) | 50 (8.2%) |
| quite | 14 (42.4%) | 128 (22.2%) | 142 (23.2%) |
| much | 6 (18.2%) | 144 (24.9%) | 150 (24.5%) |
| very much | 6 (18.2%) | 234 (40.5%) | 240 (39.3%) |
| Q13—With respect to face-to-face activities experienced * | Tutor (n. 26) | Student-teacher (n. 473) | Tot. 499 |
| much less effective | 0 | 28 (6%) | 28 (5.6%) |
| less effective | 1 (3%) | 44 (9.3%) | 45 (9%) |
| equally effective | 11 (33.3%) | 296 (62.5%) | 307 (61.5%) |
| more effective | 12 (36.3%) | 84 (17.7%) | 96 (19.2%) |
| much more effective | 2 (6%) | 21 (4.4%) | 23 (4.6%) |
| Q14—Strength | Tutor (n. 33) | Student-teacher (n. 578) | Tot. 611 |
| study time support | 8 (24.2%) | 238 (41.2%) | 246 (40%) |
| distance overcoming | 16 (48.5%) | 203 (35.1%) | 219 (35.8%) |
| greater content sharing | 5 (15.1%) | 78 (13.5%) | 83 (13.6%) |
| group-work support | 4 (12.1%) | 23 (4%) | 27 (4.4%) |
| other | 0 (0%) | 18 (3%) | 18 (2.9%) |
| no strength | 0 (0%) | 18 (3%) | 18 (2.9%) |
| Q15—Weakness | Tutor (n. 33) | Student-teacher (n. 578) | Tot. 611 |
| no weakness | 7 (21.2%) | 214 (37%) | 221 (36.2%) |
| poor interaction: stud.-stud. | 7 (21.2%) | 174 (30%) | 181 (29.6%) |
| poor interaction: stud.-tutor | 16 (48.5%) | 131 (22.6%) | 147 (24.1%) |
| other | 2 (6%) | 38 (6.5%) | 40 (6.5%) |
| content difficulty | 1 (3%) | 21 (3.6%) | 22 (3.6%) |

* From May 2021 to April 2023.

As regards the perceptions of effectiveness on online practicum activities (Q13), the table shows a general positive disposition (39.3%), especially from student-teachers (very much—40.5%), and less from tutors (quite—42.4%).

Based on the comparison with the same activities provided face to face (Q14), the population that participated in online practicum activities considers them equally effective (61.5%), above all the student-teachers (62.5%), less the tutors (33.3%).

The strengths of online practicum activities (Q18) recognized by both, tutors and student-teachers, are the support with respect to study times (40%) and the possibility of overcoming geographical distances (35.8%). The latter is especially prevalent among tutors (48.5%).

As regards the weaknesses of online practicum activities (Q19), in general 36% does not find them, while 29.6% and 24.1% report the limited possibility of interaction between student-student and student-tutor, respectively, the latter specifically for tutors (48.5%).

In reference to the second research question—related to the types of practicum activities provided online in the year 2021/22—the previous data are comparable with what emerges from the qualitative data analysis carried out on question n. F1 and focused on the report of personal experiences. Text coding was carried out on the open question (a textual corpus of n. 236 strings), first without (see Table 7), then with reference to respondents (tutors and student-teachers) (see Table 8).

**Table 7.** Categories and codes emerging from text corpora, with n. occurrences.

| Core Category | Axial Coding | Open Coding | Sub-Codes | Excerpts |
|---|---|---|---|---|
| types of activities (n. 236) | function respect to the path (n. 135) | meeting between students and tutors (n. 42) | | 'the online mode has proved to be an opportunity to "engage" school tutors and female students in shared and functional moments for each one's professional/personal activity' |
| | | thematic insights (n. 54) | sharing of materials | 'the online mode made it possible to share materials (images, videos, documents) otherwise impossible in person' |
| | | | innovative teaching content | 'multiple online applications that allowed for greater use and sharing of hitherto unknown resources' 'with the online mode we have much more material at our disposal to work with' |
| | | listening to testimonials (n. 39) | | 'it was an activity that took place as it could have been done in person' |
| | form of organization (n. 101) | small group (n. 53) | more interaction | 'paradoxically online is more about sharing ideas because it is possible to exchange materials' |
| | | | random assignment | '(thanks to) the possibility of being sorted "randomly" into small groups, I was able to talk to people with whom I otherwise would not have been able' |
| | | plenary + individual work (n. 48) | | 'have allowed the sharing of many multimedia contents' |

**Table 8.** Categories and codes related to respondents (tutor and student-teachers), with n. occurrences.

| Respondents | Axial Coding | Open Coding | Sub-Codes |
|---|---|---|---|
| Tutor (n. 38) | function respect to the path (n. 26) | meeting between students and tutors (n. 12) | |
| | | thematic insights (n. 14) | sharing of materials |
| | form of organization (n. 12) | small group (n. 12) | |
| Student-teachers (n. 198) | function respect to the path (n. 109) | thematic insights (n. 70) | sharing of materials |
| | | | innovative teaching content |
| | | listening to testimonials (n. 39) | |
| | form of organization (n. 89) | small group (n. 53) | more interaction |
| | | | random assignment |

The qualitative data in Table 7 show noteworthy aspects. The main emerging category concerns the types of activities (n. 236), diversified on the basis of the function performed with respect to the entire training path (n. 135) and the forms of organization (n. 101).

Compared to the first axial code, three open codes emerge: meetings between students and tutors (n. 42), the study of specific topics (n. 54) and listening to testimonials (n. 39). It is worth noting that the open code 'thematic insights' is in turn diversified into sub-codes linked to peculiar aspects, such as the possibility of sharing material and exploring innovative educational solutions. Compared to the second axial code, two open codes emerge: the organization for small groups (n. 53) and for plenary meetings that precede moments of individual work (n. 48). We note that even the open code 'organization for small groups' is internally divided into two sub-codes which express the qualitative values, i.e., the possibility of greater interaction between the members of the group and of getting to know new companions thanks to the random assignment function available on the learning management platform.

Table 8 shows the different coding of categories by tutors and student-teachers. Compared to the functions of the activities carried out online, the tutors focus more on the possibility of jointly meeting student-teachers and school tutors and on sharing materials, in the context of thematic insights; the student-teachers, on the other hand, focus both on in-depth study of the topics—with references to both sub-categories (sharing of materials and innovation of teaching contents)—and on the forms of organization, i.e., on the benefits of group work (more interaction and random assignment).

Participation, both by tutors and student-teachers, has progressively increased and has remained high in the transition from the pandemic phase to the post-pandemic phase (see Table 4). In the post-pandemic period (from May 2021 to today) only part of the reflection activities on the practicum continued to be provided online for about 12 h a year. These activities are, moreover, organized in plenary meetings (see Table 5). As regards the reflection activities on the practicum conducted at the university from May 2021 until April 2021, both student-teachers and tutors expressed a good perception of effectiveness, in general, and equally good compared to the activities carried out in presence. The strengths of these online activities were finding study time support, above all for the student-teachers, and the overcoming of spatial distances, above all for the tutors (see Table 6). The weaknesses highlighted were few and associated, above all on the part of the tutors, with the lack of interaction between students and tutors. The coding of qualitative data has also made it possible to understand even better some representations regarding the usefulness of online practicum activities: by the student-teachers, the possibility of making

in-depth thematic insights, thanks to the sharing of materials, and interacting more with the members of the work, and thanks to the exchange of materials and the 'random' formation of groups; by the tutors, the same possibility, thanks also to the sharing of materials and meeting with the school mentors.

## 4. Discussion, Limitation and Conclusions

The study carried out in the 'Primary Education Sciences' degree course had the objective of learning about the perceptions and representations of tutors and student-teachers with respect to the methods of providing practicum within the degree course before, during and after the pandemic phase—see Table 1. The study made it possible, in general, to learn about online practicum experiences—participation, type of online practicum activities and perception of effectiveness—and the related representations by student-teachers and tutors [25,26]. Above all, it was possible to know if these perceptions had changed in the delicate transition from the emergency phase to the post-pandemic one.

Unlike the study by Brina and Psoni [25] on post-pandemic online practicum, the present study investigated its effectiveness on the basis of the perceptions and representations of student-teachers and tutors. In essence, the same availability for flexibility and adaptability emerges, above all, the perception that online activities can encourage interactions between students, and, moreover, can offer an opportunity to exchange ideas and materials and, for this reason, support the learning.

The study was conducted on the same degree course, provided in two universities with different numbers of students (UniBa is large, UniBg is medium-sized) and geographical area (UniBa in the south, UniBg in the north of Italy) but having a very similarly path (see Table 1). However, it would be advisable to start a comparative study with other universities with further characteristics to verify possible influences on representations. At the moment, the detectable aspect emerging from the data concerns the population of tutors and previous online practicum experience. Indeed, from a descriptive point of view, it is possible to note that having experienced online practicum activities influences the perception of effectiveness of tutors in a slightly positive way (see Q13, 18.2% and Q14 36.3%) and the perception of student-teachers in a slightly negative way (see Q13, 24.9% and Q14 17.7%). A correlational statistical study will be conducted on this aspect in the next phase of analysis.

As an essential activity for teacher training, during the COVID-19 pandemic, it was necessary to find innovative solutions to implement the practicum and allow student-teachers to continue practicing in the school context. Experiments carried out prior to the pandemic phase were taken into account, whereas in e-learning environments, the learning activities usually carried out face to face were adapted for an online mode [10,11].

The online practical activities carried out on an experimental basis before the pandemic phase therefore suggested possible temporary solutions for carrying out remote practice which, in the post-pandemic phase, were confirmed or, in turn, adapted in a hybrid way. It is possible to argue, then, that the pandemic has somehow accelerated the push towards innovation in the forms of organization/provision of practical activities—no longer carried out exclusively in person but increasingly in a hybrid mode [13] and at increased levels [14].

Already during the pandemic and even more so in the post-pandemic phase, some teacher training research has investigated the effectiveness of adapting the online practicum from multiple perspectives: in terms of the general balance between the challenges and opportunities posed [12,19] and as positive effects found on the transversal skills of the student-teachers involved [25]. Some studies, in particular, have focused on student-teachers' perception of effectiveness, linked to the feeling of being able to better reconcile working time [27] or of better collaborating flexibly with others [16]. The present study continues the theme of the perception of effectiveness of the online practicum from the point of view not only of the student-teachers but also of the tutors: effectiveness is associated, on an organizational level, with overcoming spatial distance (especially from the perspective of the tutors) and, in terms of learning processes, the in-depth study of the

themes thanks to the greater exchange of materials (from the perspective of the student-teachers). The study contributes by making known the point of view of student-teachers and tutors on hybrid and online forms of practicum delivery and highlighting the need for relationships and learning support. These are aspects that those responsible for training courses should take into due consideration to better respond to the people involved in various capacities in teaching training courses and that researchers should investigate to gain a closer understanding of the complex dynamics of teacher training.

**Author Contributions:** L.P. is author of paragraphs 1 and 2; L.S.A. is author of paragraphs 3 and 4. All authors have read and agreed to the published version of the manuscript.

**Funding:** This research received no external funding.

**Institutional Review Board Statement:** The study did not require ethical approval. The data was collected and analyzed anonymously.

**Informed Consent Statement:** Informed consent was obtained from all subjects involved in the study.

**Data Availability Statement:** The data presented in this study are available on request from the corresponding author.

**Conflicts of Interest:** The authors declare no conflict of interest.

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
