# Peer review of "Online Practicum Effectiveness: Investigation on Tutors’ and Student-Teachers’ Perceptions Pre/Post-Pandemic"

_education, doi:10.3390/educsci13121191_

Round 1
Reviewer 1 Report
Comments and Suggestions for Authors
Contribution is original and interesting by proposing a model that combines live experience with online experience. The methodological approach is coherent with the theoretical framework and data analysis. Relevant references.
Author Response
Thanks for your comments. I add the revisited file

Reviewer 2 Report
Comments and Suggestions for Authors
Thank you for allowing me to review your manuscript with Education Science-2565647. Your topic is worthy, and you did a great job here. However, I have several concerns for you, and I hope they are helpful.
1) The abstract is vague, and it needs improvement. For example, you use the mixed research design, but in the main manuscript, I wonder how you use the mixed research design. I wonder about using mixed ways to collect data, which is a mixed research design. Also, the findings are very vague as well. Therefore, the abstract needs improvement thoroughly.
2) The whole manuscript needs to be revised thoroughly—for example, your topic talks about the perspectives of effective online practicum from tutors and student teachers. However, the introduction did not focus on the areas you intend to align with the topic. This manuscript does not align with your topic, and you need to have more data to support it. Also, you mentioned how COVID-19 impacts online practicum in the introduction, but I could not see how this is related to your topic with a strong argumentation.
3) The purpose of this study, research questions, and research design need further improvement since your study is a mixed design related to online practicum.
4) The mixed research design needs to be revised and improved. The current one is not strong enough and is an illogical presentation.
5) Finding the presentation is not effective, as well as the conclusion. The reason may be caused by research design. Therefore, I suggest revising it thoroughly.
Comments on the Quality of English Language
Proofreading is needed.
Author Response
The abstract has been better focused
The topic investigated is in line with the premises and it is explained in paragraph 1. Results are made more explicit in response to the research question and summarized at the and of the paragraph 3. Results are made more explicit. Some passage are revised at linguistic level Conclusions were improved as suggested

Reviewer 3 Report
Comments and Suggestions for Authors
This is an interesting and carefully reported study which investigates the perceptions of student-teachers and their tutors engaged in the practicum aspects of BA Primary Science degree in two Universities in Italy.
Regarding the title of the study, I would encourage the authors to consider revisiting the title and replacing the words "Really" and "Effective" with adjectives and verbs that are a little less causal in essence and these could be construed as weakening interest the study and limiting the potential readership of what is otherwise a well-written article.
Page 2 line 44 the authors might consider rephrasing this sentence as follows:
“ … due to differences in what they live in relation to their direct experience and what they have been taught in their programme if initial teacher education.”
The content of the study is succinctly described and clearly contextualized. The literature review draws upon an appropriate range of relevant and recent peer-reviewed, published literature and the references cited are relevant to the research. The mixed-methods design is clearly stated including a description of a research population of 33 tutors 578 student-teachers and regarding practical supervision offered to student-teachers online. Arguments are coherent, balanced and convincing. Results are presented with commendable care and transparency. Conclusions are clearly supported by the results presented in the article and reported with criticality and reference to relevant literature.
Comments on the Quality of English Language
Only minor editing of English language is required. Please see my suggested amendment on page 2 Line 44
Author Response
The title has been changed as suggested
The abstract has been better focused
The sentence in the introduction has been changed
Results are made more explicit in response to the research question and summarized at the and of the paragraph 3.
The reference to the mixed design was eliminated and the mixed nature of the data collection tools was made explicit
Results are made more explicit.
Some passage are revised at linguistic level
Conclusions were improved as suggested

Round 2
Reviewer 2 Report
Comments and Suggestions for Authors
Thank you for giving me this chance to re-review your manuscript, which you did a great job.